# Advancing Mental Health and Equity Through Infant and Early Childhood Mental Health Consultation

**DOI:** 10.3390/healthcare13050545

**Published:** 2025-03-03

**Authors:** Jennifer Drake-Croft, Amittia Parker, Lauren Rabinovitz, Rachel Brady, Neal Horen

**Affiliations:** Thrive Center for Children, Families, and Communities, Georgetown University, Washington, DC 20007, USA; amittia.parker@georgetown.edu (A.P.); lar99@georgetown.edu (L.R.); rab9@georgetown.edu (R.B.); horenn@georgetown.edu (N.H.)

**Keywords:** infant and early childhood mental health, equity, infant and early childhood mental health consultation, early intervention

## Abstract

Early intervention services are a critical support for young children experiencing developmental delays and disabilities. Due to myriad negative social and economic conditions, some infants and young children, namely Black, Indigenous, and other children of color, as well as those experiencing poverty, are at greater risk of experiencing a developmental delay or disability and experiencing issues of access to needed services and supports within and beyond early intervention programs. Due to these systemic issues, these infants and young children are more likely to have caregivers experiencing mental health concerns and issues of access to services and supports. Early childhood serving programs are faced with meeting the behavioral health needs of families experiencing cumulative vulnerabilities. Some early intervention (EI) programs are partnering with infant and early childhood mental health (IECMH) providers to meet mental health needs. IECMH consultation (IECMHC) is a multi-level support that aims to build the capacity of early childhood programs to meet the needs of young children, families, caregivers, and staff. IECMHC has an intentional focus on promoting and ensuring equity, specifically more equitable systems. It focuses on addressing inequities impacting young children and their caregivers, thus strengthening these essential collaborations. This paper highlights research demonstrating the importance and collective power of IECMHC in early intervention programs to advance behavioral health and equity.

## 1. Introduction

In the first three years of life, young children form close relationships and learn, grow, and develop rapidly in the context of their family, culture, and community [1,2]. Early intervention (EI), also known as early childhood intervention, is a critical support and service for families with young children at risk of or experiencing a developmental delay or disability [3,4]. Providing services to young children and their families within their natural environments is widely recognized as a global best practice and is mandated in countries such as the United States [5,6].

Unfortunately, disparities or differences in risk factors, needs, and inequitable access to services and outcomes persist. Due to negative social and economic conditions or social determinants of health, marginalized children, including children of color and those living in poverty, are more likely to be exposed to early life stressors and adversities, as well as being exposed to environmental toxins in utero and the early years, placing them at risk of experiencing developmental delays and disabilities, special health care needs, and social–emotional–behavioral (SEB) concerns [5,7].

Infant and early childhood mental health consultation (IECMHC) is a multi-level intervention that aims to build the capacity of early childhood programs to meet the needs of young children, their families, other caregivers, and staff within their programs [8,9]. IECMHC has an intentional focus on promoting social–emotional development and positive relationships and ensuring equity [10]. The purpose of this paper is to highlight *why* early intervention and IECMHC programs should and can work together to support families with young children, and how they can help reduce disparities and advance equity.

This synthesis employed a systematic approach to identify, review, and integrate relevant research on EI and IECMH. Peer-reviewed journal articles, policy reports, and gray literature were included to ensure comprehensive coverage of the field. Sources were identified through keyword searches in academic databases and were screened for relevance based on predefined inclusion criteria. Key themes, trends, and gaps in the literature were extracted and analyzed to provide a cohesive narrative and critical evaluation of the current state of knowledge, as well as to identify implications for practice, policy, and future research.

### 1.1. Early Intervention

In the United States, the Individuals with Disabilities Education Act (IDEIA, 2004) mandates that states and localities provide services and supports aimed at promoting child development and well-being while reducing the need for future special education services. The Section of this law, Part C, specifically defines and describes early intervention services, designed to support, from birth to age three, eligible infants and toddlers with developmental delays or disabilities and their families [11]. The goal is to foster young children’s development and empower their families to meet their needs through diverse therapeutic and educational interventions provided in natural environments. Early intervention, often referred to outside the United States as early childhood intervention (ECI), can include services such as speech therapy, physical therapy, and occupational therapy, as well as family education and support.

By equipping families and caregivers with tailored tools to address developmental challenges when the brain is most plastic, developmental delays can be mitigated, improving lifelong outcomes [5,12]. Equipping families and caregivers includes building responsive relationships among caregivers and children [1]. Services are tailored to the individual needs of each child and family and are delivered within natural learning environments, including the home and community settings, such as schools or early education and care spaces, to maximize effectiveness. Around the globe, countries have different ways of providing services and supports for infants and toddlers, but all include some aspects of the natural environment and family-centered practices [13]. For this paper, EI is defined as family-centered care interventions including those that build relationships as needed to support young children at risk of or experiencing disabilities.

### 1.2. The Intersection of Infant and Early Childhood Mental Health and Early Intervention

Infant and early childhood mental health (IECMH), also referred to as social and emotional development, is the developing capacity of the infant/young child to form close and secure relationships; experience, manage, and express a full range of emotions; and explore the environment and learn—all in the context of family, community, and culture [2,14]. Early relational health with caregivers plays a crucial role in fostering the development of IECMH. Attuned, responsive caregiving—characterized by consistent ‘serve and return’ interactions, in which caregivers and children engage in reciprocal exchanges—shapes the developing brain architecture, thereby influencing all developmental domains [1].

Developmental delays are often caused by a complex interplay of factors, with some arising solely from genetic conditions, while others may develop as a result of external factors such as environmental toxins or parental neglect [15]. Regardless of the underlying causes of developmental delays, the presence of safe, stable, and nurturing relationships and environments plays a major role in mitigating these delays [16]. As IECMH is integral to all areas of development, addressing it is a fundamental aspect of any infant or early childhood support, including EI.

Children with social–emotional–behavioral (SEB) concerns are more likely to experience developmental issues such as speech and language delay, attention and hyperactivity differences, and co-occurring physical health concerns such as asthma [3,17]. Furthermore, in the United States, approximately 32% of children under the age of five years old demonstrate significant SEB delays [3]. The literature often addresses physical health, developmental health, and SEB health separately. However, many families with young children face a range of overlapping vulnerabilities and needs as they navigate EI and behavioral health systems of care. Although data indicate that children receiving EI services for SEB delays may have unmet needs, EI programs often fail to consistently address SEB concerns, likely due to the variability in training, support, and perceived competence among EI professionals [3].

#### 1.2.1. Caregiver Stress

Infant and early childhood SEB delays and concerns impact the young child and their health, well-being, and functioning within their environment, significantly impacting the entire family system. Caregivers of children experiencing social–emotional delays report high levels of stress, employment, and financial concerns, as well as physical and mental health concerns [17,18,19]. Caregiver stress can undermine the quality of early relational health, which, in turn, negatively affects IECMH and the development of executive functioning [20,21]. Caregivers express a need for mental health support, practical guidance to manage their children’s behavior, and tangible assistance to meet their families’ needs, such as help with housing, finding a physician, or securing transportation [18,22,23]. There is a need for interventions that are culturally responsive, comprehensive, and tailored to address the needs of both young children and their families. Multigenerational interventions that support both children and adults are beneficial to caregivers, helping to reduce stress, improve attitudes and behaviors, and enhance caregiver mental health [18,24,25]. Although EI programs are family-centered interventions designed to support families with young children at risk of or experiencing developmental delays and disabilities, data suggest that EI providers often struggle to meet the SEB needs of these children [25].

#### 1.2.2. Disparity Among Children with Developmental Delays

The same socioeconomic factors that increase the likelihood of developmental delays (e.g., limited financial resources or experiencing early adversities) are found disproportionately among marginalized populations, including communities of color in the United States [26,27]. From conception, these conditions inequitably compromise maternal health, fetal development, and birth outcomes, resulting in compromised infant and early childhood development and poorer outcomes across the lifespan [28,29,30,31]. These systemic and structural factors further inequitable access to EI services in the United States [32,33,34]. Even when historically marginalized children with disabilities are identified, there is a disproportionate delay in receiving needed EI services at the period known to be critical to mitigate risks and promote growth and development [35,36].

Throughout the world, children with disabilities are more likely to experience child abuse and neglect and reduced access to services and supports, creating a vicious, disruptive cycle of compounding factors across the lifespan that lead to disease, disability, and poor social outcomes [37]. The Global Burden of Disease (GBD) Study estimated that 53 million children under 5 years globally experience developmental delays or disabilities, including epilepsy, sensory impairments, and autism spectrum disorder, and that 95% of those children reside in low- and middle-income countries [38]. The high rates of delay and disability in children from birth to age five in low- and middle-income countries, where fewer families have access to effective family-centered early childhood interventions, highlight the link between economic access and child outcomes [38,39].

## 2. Early Interventionists Lack Sufficient Professional Support in IECMH

Advances in developmental science have heightened the emphasis in early intervention on addressing SEB needs while promoting family-centered practices in natural environments [40]. In recognition of the vital role primary caregiving relationships play in cognitive and social–emotional development, EI principles emphasize family-centered practices, where early interventionists collaborate with families as partners, consultants, and problem solvers rather than positioning themselves as experts imparting knowledge [4,41]. Professionals apply these best practices by carefully considering each family’s unique background, values, and available support systems. Using family-centered practices allows professionals to seamlessly incorporate intervention strategies into everyday life and family interactions, ultimately building trust with families, enhancing caregiver confidence and competence, and embodying IECMH principles [40,42,43].

The complex factors impacting children in need of EI highlight the critical need to equip the EI workforce with a holistic understanding of both the child and family. Equipping the EI workforce includes addressing the challenges of IECMH and developmental needs, particularly for children from historically marginalized communities. The awareness and recognition of a professional’s own beliefs, values, and biases are essential to family-centered practice. Translating theory into practice requires sufficient professional development and ongoing opportunities to apply knowledge and build skills, yet there are no standard practices to support application [41,42]. EI practitioners largely do not receive training outside of their area of expertise (e.g., occupational therapy) and do not consistently receive supervisory or other forms of professional support [42]. Furthermore, in the United States, inconsistent requirements for certification and licensure, a lack of professional development opportunities, and insufficient state infrastructure to recruit, train, support, and sustain a qualified workforce compromise the delivery of high-quality EI services [44].

## 3. Infant and Early Childhood Mental Health Consultation

Given the prevalence of young children experiencing SEB concerns and the impact of these concerns on lifelong learning, behavior, and health, a multi-level equity-centered intervention like infant and early childhood mental health consultation (IECMHC) is an underutilized support [19,45,46]. This multi-level intervention is delivered by mental health consultants trained in early childhood development and the core principles of infant and early childhood mental health consultation (IECMHC). These principles include relationship-based care, collaboration, individualized support, cultural and linguistic responsiveness, knowledge of child development, evidence-based practices provided in natural community settings, and a continuum of strategies ranging from promotion to intervention [47,48].

EI programs that collaborate with IECMH-informed professionals have experienced some success; however, in many places, these programs operate independently, without realizing the power of their partnership and the value they can add to each other and the families they serve [49]. EI and IECMHC work in harmony to complement and strengthen one another [50]. A growing body of literature on evidence-based home visiting programs, similar to EI, attract professionals from diverse backgrounds and leverage IECMHC to strengthen provider capacity and equip them with the skills needed to support families facing behavioral health challenges [51].

Adequately supporting children who exhibit challenging behaviors, are at risk of expulsion from early care and education programs, or have developmental disabilities and delays requires cross-disciplinary collaboration to meet their needs and support their families effectively [52]. Building a collaborative relationship between early interventionists and IECMH consultants is likely to benefit EI providers and programs, similarly to the ways in which IECMHC has been helpful to early childhood early child development center staff, home visitors, and child welfare workers [8]. In addition to building the capacity of providers to respond to a child’s SEB needs, consultation improves staff wellness, increases self-efficacy, and reduces job stress and turnover [9,25,50].

Although there is recognition of the benefits of strong EI and IECMHC partnerships [50], there is a limited body of literature exploring this critical collaboration. Little is known about how a collaboration between EI and IECMHC can address disparities in early childhood and advance equity.

### 3.1. Integrating Infant and Early Childhood Mental Health Consultation into Early Intervention to Advance Equity

The theory of change (ToC) for IECMHC developed by the Center of Excellence (CoE) for IECMHC (Figure 1) posits that the formation of a relationship between consultant and consultee that is trusting and collaborative leads to the direct impact of a consultee’s increased capacity to promote IECMH, which, in turn, indirectly improves programmatic outcomes, child and family outcomes, and a reduction in disparities [53]. The special relationship between the consultant and the consultee and the activities of consultation influence each other reciprocally, and both are shaped by the consultant’s participation in reflective supervision [54,55].

As EI is a newer field for IECMHC implementation (at least documented, not necessarily in practice), the ToC helps explain the process by which IECMHC in EI can better help young children with delays and disabilities, EI professionals, and their families, have opportunities to collectively advance equity by actively addressing systemic barriers and biases within early childhood settings (See Figure 1). Smith and colleagues (2020) described IECMHC as being available in EI (based on state self-report), but it is likely not IECMHC as defined by the CoE nor consistent with the ToC. For example, having someone identified as a consultant who provides an expert opinion on the eligibility process of a child, but is not involved in any ongoing capacity, is not consistent with the ToC. IECMHC is a dynamic process between the consultant and consultee who together and over time increase the capacity of the consultee to support children and families. Having experts in mental health as part of EI eligibility or providing external evaluations are an important component of EI services; it, however, should not be confused with or equated with what the CoE describes as an IECMH consultant.

Regardless of setting, IECMHC will always have a consultant and consultee engaging in a relationship to support the direct and indirect outcomes of consultation, as noted in Figure 1. What can vary widely by setting and by individual IECMHC program is the consultee and the specific IECMHC activities. Currently, the CoE has adopted a set of competencies, including one focused on equity, as outlined in the ToC. However, there is no national accreditation for IECMH consultants or programs, leading to variability in practice. Like IECMHC, there is variability in how EI is implemented within the United States [56]. While EI is a federally funded program, states have flexibility in how services are administered and distributed, as long as it follows the federal regulations of the program. The terminology of the different service providers varies, as does the eligibility process.

In each element of the ToC, there are equity considerations that should inspire questions and attention to critical equity issues [53]. The following section will describe each element of the ToC and how it can be applied within an EI program.

#### 3.1.1. Participant Characteristics

Participant characteristics are the first element of the ToC and a newer area of research in the literature on IECMHC. Studies have begun to explore the relation between participant characteristics such as intersecting race, gender, age, disability, socio-economic class, and well-being outcomes such as stress and mental health concerns (e.g., anxiety or depression). Other background characteristics such as education and work-related training and experiences (job satisfaction and burnout) are also important to consider.

In the application of IECMHC in EI as a program and in an embedded setting, the consultee is the EI provider or program. The EI backgrounds such as their personal attributes (race, gender, stress, mental health, and burnout) and professional experiences (education, training, and work experiences) in their role are factors that might influence their work and also their readiness for consultation and experience with consultation. The consultant’s personal attributes (race, gender, stress, mental health, and burnout) and professional experiences (education, training, and work experiences) in their role are also important factors that might influence their ability to embody the consultative stance and build strong and impactful relationships with children, families, and staff.

#### 3.1.2. Reflective Supervision

Reflective supervision is another element within the ToC that is an ongoing support for the consultant in order to facilitate high-quality consultation for the consultee. Reflective supervision is experienced as supportive and beneficial among practitioners within infant and early childhood settings and is an important part of infant and early childhood mental health practice [57,58]. In their own reflective space, the consultant can engage in a parallel process to examine how systemic oppression might influence their work with diverse children experiencing delays and disabilities.

#### 3.1.3. Engaging in IECMHC

This component of the ToC highlights the continuous interactions between the consultant and the consultee, emphasizing the consultant’s actions and activities, and the development of a strong consultative alliance or relationship. IECMHC is unique in its individualized approach to meeting the specific needs of each program. A growing body of literature documents the core equity-focused activities in IECMHC and their role in fostering the consultative alliance and influencing outcomes [59]. The activities and the relationship mutually reinforce each other, with research suggesting that reflective supervision, through a parallel process, influences both the consultant and the consultee in how they approach their work. In the context of EI, these findings suggest that engaging with IECMHC enables EI providers to establish strong consultative partnerships and actively contribute to IECMHC initiatives. Participation in IECMHC fosters better relationships with children and families, improves child behavior, and enhances the caregiving environment for children, families, and staff.

#### 3.1.4. Impact of IECMHC: Direct and Indirect Effects

IECMHC is an indirect intervention that has long-term effects on children, families, and programs [9,10,60,61]. Change unfolds as a dynamic process between the consultee (in this case, the EI provider(s)) and the consultant. Shifts in knowledge, perceptions, emotions, relationships, and behavior enhance understanding, empathy, and the ability to respond sensitively to the needs of children, families, and staff. The dynamic process of IECMHC ultimately benefits EI providers, programs, early childhood education partners, and the children and families they serve.

The existing literature on IECMHC largely focuses on early childhood settings, particularly within ECE contexts (center-based, family childcare, and home-based), rather than in EI programs. In EI work, we are applying established knowledge to a new context to inspire collaboration and drive meaningful change for improved outcomes.

We anticipate that integrating IECMHC into the EI program will enhance EI staff’s understanding of social–emotional–behavioral development, mental health in children and families, and staff well-being. The integration of IECMHC and EI is also expected to improve their skills in engaging effectively with children, families, and colleagues. Through consultative relationships and IECMH activities, EI staff can build greater capacity to work with caregivers and children in culturally responsive, reflective, and supportive ways, aligning more closely with the intended outcomes for children and families. Additionally, we expect similar benefits in staff wellness and relationships to those observed in studies involving early childhood educators.

### 3.2. System-Level Vignette: State of Alabama, United States

Quote from Dr. Chelsea Taylor, IECMHC Consultant: “I think a lot about ecological theory [Bronfenbrenner] because we always want to hold the child in the center, and I feel as though EI is already providing that support to the family, to the children, to the child, in that household. And then we [IECMH consultants] are the outer layer surrounding EI. So, we are there to support EI in their efforts to support the family, and you can never have too much support.”

Alabama serves as an example of a statewide effort to implement IECMHC in EI. In 2014, the Alabama Department of Mental Health (ADMH) received a Project LAUNCH grant from SAMHSA to implement activities in Tuscaloosa County, home to the University of Alabama, while simultaneously building a statewide infrastructure to develop the IECMH System of Care. As part of the grant, the first two IECMHCs were hired in the state to work within the two EI programs in Tuscaloosa County, as well as in other early childhood sectors. From here, a state-level IECMH Coordinator position was created at the Alabama Department of Early Childhood Education (ADECE), with 50% of the funding provided by the Project LAUNCH grant. That position is now called the State IECMH Partnership Director and continues to be funded by ADECE and ADMH. The grant ended in 2019 but was the beginning of exciting IECMHC developments in the state, including the following:Ten statewide IECMHC positions to support licensed childcare centers in 2019.Ten additional statewide IECMHC positions created by ADMH, amidst the COVID-19 crisis in 2020, through educational funds to support a variety of early childhood programs including EI.Additionally, in 2020, ADMH created an IECMH Services Coordinator position to provide administrative supervision to the consultants and to help coordinate the state’s professional development for IECMH workforce capacity building, as well as a State Reflective Practice Coordinator to provide reflective supervision to the consultants.IECMHC support in EI began as a pilot at the end of 2020 and is now offered statewide.

The IECMHC program in Alabama is delivered through a tiered system. Tier 1 is Training and Education for the EI workforce; Tier 2 is providing Program and Team Support through Reflective/Case Consultation; and Tier 3 is child- and family (of child receiving EI)-specific consultation. The program provides a Tier 4 option when deemed beneficial, shifting from consultation to direct treatment for the child/parent dyad. This complement to the consultation services can be provided by all consultants who are licensed MH clinicians. When Tier 3 or Tier 4 is needed, they are listed on the child’s IFSP as Non-EI Service. None of the IECMHC tier services are paid for by Part C dollars, nor are they charged to the family or the EI programs. The IECMHC leadership team is currently working with the Alabama EI System leaders to increase the utilization of the services [62].

## 4. Conclusions/Call to Action

There are numerous behavioral health interventions that EI programs can explore to address the social–emotional and mental health needs of families with infants and toddlers facing developmental delays and disabilities [49,50]. EI and IECMHC share significant alignment in their approaches to working with young children and families, particularly in emphasizing the importance of partnering with and supporting the adults in children’s lives. Both also uphold shared values and beliefs in promoting equity to address disparities in access and outcomes, which disproportionately affect children of color, children from low-income backgrounds, and those experiencing early adversities.

A natural complement in the strengths between needs among EI and IECMHC is also present. For example, EI providers have deep knowledge, skills, and experiences supporting the developmental needs of infants and toddlers, and IECMHCs have knowledge and skills related to social–emotional development, trauma-informed approaches, and the mental health needs of caregivers. EI providers and IECMHCs report needing support in the area in which the other has expertise, and both are interested in building the capacity of adults in the lives of young children (e.g., parents/caregivers, educators, or other providers).

IECMHC predicts positive child outcomes, as greater engagement with a consultant is associated with improvements in challenging behaviors and social skills, as well as reductions in provider stress, increased job satisfaction, enhanced self-efficacy, improved service quality, and a lower risk of early childhood expulsion [59]. IECMHC is a promising, evidence-based practice, and the proposed processes and outcomes for strengthening and embedding collaboration between EI and IECMHC have the potential to enhance services and promote equity. The authors provide recommendations for research and practice to build and sustain strong collaborations.

### 4.1. Recommendations for Practice

Build strong partnerships and collaborations with a variety of mental health supports and services, especially those who specialize in infant and early childhood mental health.Embed IECMHC into EI programs: Integrate IECMHC into EI programs to provide multi-level mental health intervention services within the program in ways that benefit children, families, EI providers, EI programs, and their communities.Facilitate joint training: The nature of the work can be stressful for EI providers and IECMHCs, and joint training on infant and early childhood mental health, early intervention, and advancing equity might help to bridge knowledge, understanding, and skills gaps, as well as facilitating connections for collaboration.

### 4.2. Recommendations for Research

Implementation challenges have persisted in integrating two complex and distinct systems. These systems face numerous obstacles to collaboration, including differences in funding structures, policies, communication methods, and health record systems. A tailored, cross-disciplinary, and extended training process may offer a potential solution by fostering connections, improving collaborative communication, and facilitating cross-training to bridge the gap between them. The following research recommendations aim to advance understanding of how IECMHC can benefit EI professionals, as well as the caregivers and children they support.

Ensure that screening and assessments identify risks for developmental delays and disabilities including social–emotional–behavioral delays, as well as early childhood adversities including poverty, parental mental health and substance use history, experiences with domestic or community violence, and history of early childhood suspension or expulsion. In addition, explore issues of access or barriers to needed and preferred services and supports.Systematically explore the state of collaboration between EI and IECMHC. There is much to learn about what is effective, for whom it works, and how the expected or theorized outcomes are achieved. Examine the reach and impact of IECMHC and equity-focused professional development in EI programs.Apply implementation science to evaluate collaborations between EI and IECMHC. Additional research is needed to explore critical implementation outcomes—such as reach, adoption, fidelity, impact/effectiveness, and sustainability—in efforts that involve collaborative community partnerships between EI and IECMHC. This includes programs that embed IECMHC services directly within existing early intervention initiatives.

In conclusion, EI and IECMHC are critical partners in helping to meet the needs of young children at risk of or experiencing developmental delays and disabilities and SEB, as well as addressing disproportionate outcomes for marginalized populations. A collective effort is both timely and essential, emphasizing how EI and IECMHC professionals intersect and can collaborate to improve services for families while reducing disparities and advancing equity.

## Figures and Tables

**Figure 1 healthcare-13-00545-f001:**
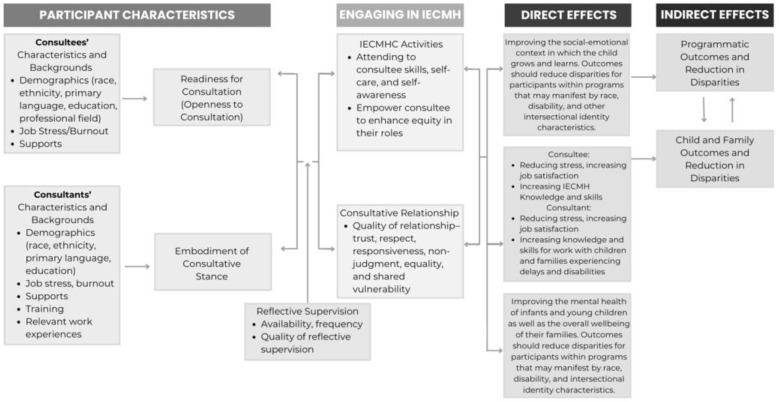
Expanded theory of change for IECMHC.

## Data Availability

Data sharing is not applicable. No new data were created or analyzed in this study.

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
