# Peer review of "Advancing Mental Health and Equity Through Infant and Early Childhood Mental Health Consultation"

_healthcare, 2025, doi:10.3390/healthcare13050545_

Round 1
Reviewer 1 Report
Comments and Suggestions for Authors
Thank you for the opportunity to review this interesting article. The writing is clear and concise. The overall concept of using IECMHC to advance equity in EI is appropriate for the special issue and of general use and interest to the field. The authors do an excellent job defining EI and IECMHC. The description of congruence between the two fields is well formed and supported by appropriate citations. I appreciated that the authors explained why infants/toddlers with disabilities and their families often need IMH supports as well as the range of reasons why EI providers need consultation to be able to give this support. The facts of inequity in access to such supports was also clearly outlined. The focus on workforce development through relationship (RSC) was clearly explained in the narrative and the ToC figure. I wanted to highlight the author’s point about the difference between having a mental health expert take part in eligibility determination and providing IECMHC over time (lines 242-244), as this is extremely important and often misunderstood.
I agree that there is limited information/research on applications of IECMHC or IMH in general to the EI space. I would suggest that there are some data on IMH and IECMHC in home visiting more generally that could apply to this article in some degree. For example, there is a 2020 report from HRSA/MCHB titled Embedding Infant and Early Childhood Mental Health Consultation in Maternal, Infant, and Early Childhood Home Visiting Programs that seems potentially relevant. Since EI and MIECHV programs also have a lot of similarities (e.g., federally funded, 2-generational, delivered in natural settings) there may be some data that could apply to EI. If the authors felt this might add to their paper, some reference could be considered around line 216.
The authors mention in the abstract that there are some EI systems that already have IECMHC embedded. I wondered if a specific example might be included right before the call to action to help the reader see how this could look.
In the recommendations section, the ones for practice seem appropriate to me. I was a little confused on the ones for research. The first one is the same for the previous section—I was unsure if that was intentional. The third one seems to be incomplete.
There are a few typos in the reference section. It looks like the first letters of some of the article titles dropped off.
Author Response
Response to Reviewer 1 Comments
|
||
1. Summary |
|
|
Thank you very much for taking the time to review this manuscript. Please find the detailed responses below and the corresponding revisions/corrections highlighted/in track changes in the re-submitted files.
|
||
3. Point-by-point response to Comments and Suggestions for Authors |
||
Comments 1: I would suggest that there are some data on IMH and IECMHC in home visiting more generally that could apply to this article in some degree. For example, there is a 2020 report from HRSA/MCHB titled Embedding Infant and Early Childhood Mental Health Consultation in Maternal, Infant, and Early Childhood Home Visiting Programs that seems potentially relevant. Since EI and MIECHV programs also have a lot of similarities (e.g., federally funded, 2-generational, delivered in natural settings) there may be some data that could apply to EI. If the authors felt this might add to their paper, some reference could be considered around line 216.
|
||
Response 1: Thank you for pointing this out. We agree with this comment. Therefore, we have added a reference on line 227.
|
||
Comments 2: The authors mention in the abstract that there are some EI systems that already have IECMHC embedded. I wondered if a specific example might be included right before the call to action to help the reader see how this could look.
Response 2: We agree with this feedback. Please see the revised manuscript with a vignette highlighting implementation work in Alabama.
|
||
Comments 3: In the recommendations section, the ones for practice seem appropriate to me. I was a little confused on the ones for research. The first one is the same for the previous section—I was unsure if that was intentional. The third one seems to be incomplete.
|
||
Comments 4: There are a few typos in the reference section. It looks like the first letters of some of the article titles dropped off.
|
||
Response 3: Thank you for this catch as well. We have made the appropriate corrections in the reference section.
|
Reviewer 2 Report
Comments and Suggestions for Authors
Dear Authors:
Thank you for the opportunity to review your revised manuscript, "Advancing Mental Health and Equity Through Infant and Early Childhood Mental Health Consultation!" Your work appears to represent a timely and valuable contribution to the field--particularly in addressing the critical intersection of early intervention services and mental health consultation through an equity lens. I really like this approach.
I've looked at both first submission and the revision, and the revised manuscript demonstrates significant improvements, particularly in its theoretical foundation and organization. Your introduction provides comprehensive background that effectively highlights the disparities faced by historically marginalized communities. The use of current literature through 2024 and relevant references strengthens your arguments considerably.
The decision to structure this as a perspective paper works well, with the Theory of Change (ToC) model providing strong theoretical grounding, in my view. The detailed exploration of how elements like reflective supervision, participant characteristics, and collaborative relationships interact is particularly insightful. I feel thgat the framework you've developed for understanding IECMHC integration into EI programs is both compelling and practical.
One notable strength of the revision is the expanded discussion of equity considerations and the benefits of IECMHC integration for both families and staff. The treatment of reflective supervision as a tool for addressing systemic oppression adds important depth to your argument. Your recommendations for both practice and research are specific and actionable, which practitioners will find valuable.
To further strengthen what is already a sufficiently strong manuscript, PLEASE consider adding:
- A brief explanation of your literature synthesis approach;
- Some concrete examples or brief case studies to help readers envision implementation; and
- A discussion of potential implementation challenges and solutions.
The lack of these elements left me dissatisfied, and I believe other readers, too, would feel the same way. The writing is clear and professional, though some longer sentences could be simplified for clarity. However, this is a minor point in what is otherwise a well-executed manuscript.
Overall, this updated version represents a significant improvement over the original submission. It offers a thoughtful and well-reasoned perspective on an important topic, with relatively clear practical implications for the field. But there is room for improvement, for sure.
With the minor additions suggested above, it will be even more valuable. Thus, I recommend another round of revisions, which should be a quick effort. Thank you for your careful attention to the revision process, and I look forward to seeing the final version!!!

Author Response
1. Summary |
|
Thank you very much for taking the time to review this manuscript. Please find the detailed responses below and the corresponding revisions/corrections highlighted/in track changes in the re-submitted files.
|
|
3. Point-by-point response to Comments and Suggestions for Authors |
|
Comments 1: A brief explanation of your literature synthesis approach;
|
|
Response 1: Thank you for pointing this out. We agree with this comment. Therefore, we have added a paragraph in the introduction detailing our approach
|
|
Comments 2: Some concrete examples or brief case studies to help readers envision implementation;
Response 2: We agree with this feedback. Please see the revised manuscript with a vignette highlighting implementation work in Alabama.
|
|
Comments 3: A discussion of potential implementation challenges and solutions.
|
|
Response 3: Given the lack of implementation, we didn’t believe we could do justice to this suggestion without reducing clarity within the paper.
|
Reviewer 3 Report
Comments and Suggestions for Authors
The piece, Advancing Mental Health and Equity Through Infant and Early Childhood Mental Health Consultation is a Perspective piece offering insight into the potential benefit for early intervention to be combined with the IECMH multi-level support. The Perspective piece was well-written and clear, beneficial for both scholars and practitioners alike.
Some minor edits to address are below.
IDEA is cited as 2004, which should be noted that for the year 2040 is the IDEIA (Improvement Act)
Oftentimes the word “this” is used, where it would be clearer for the reader to have the concept directly referred to instead of the generic word of “this.” Check throughout to help increase the clarity.
Generally, “further” is used to connect two ideas mid-sentence, whereas “Furthermore” is used to start a sentence. Fix throughout.
Page 3 line 114, why are the children’s needs not met? Additional detail would be helpful to add here.
Relationship is a term usually used to define a connection between two individuals, whereas the word relation is a term used to define the connection between two variables. There are a few instances where the word relation may be a better fit over that of relationship.
What does “advance equity” mean page 5 line 235, I think a specifier here of what this specifically means will help the reader contextualize what this looks like and shift away from an abstract goal.
Figure 1 is very helpful and well done. There are some small errors to be corrected to help with professionalism.
- Make the approach for capitalization after the bullet lists consistent
- Self-awareness, self-care generally have a dash in between the words
Indenting spacing is off in Conclusion and Call to Action.
In the Recommendations section, for practice recommendations are there any models you would suggest for collecting data on the social-emotional areas (such as specific measures or interview protocols)?
The Research recommendation for “IECMHC within early intervention programs.” is a little unclear, reading it I wonder are you suggesting program evaluations of effective IECMHC supports? Add a little more detail here to clarify what you mean .
Part of the template is still present in “author contributions”
Remove the “ in the Funding Statement
Sometimes social-emotional has a dash, sometimes not. Please evaluate for consistency throughout.
The abbreviations may be better as a table higher up in the article, or as a supplemental material.
Overall, I really liked this piece. I appreciated how the ending section gave a call for researchers and clinicians and what action oriented steps may be beneficial. That said, as a clinician, reading this paper, I would love to see resources for where to go to learn more as this was a Perspective paper.
Author Response
Response to Reviewer 3 Comments
|
||
1. Summary |
|
|
Thank you very much for taking the time to review this manuscript. Please find the detailed responses below and the corresponding revisions/corrections highlighted/in track changes in the re-submitted files.
|
||
3. Point-by-point response to Comments and Suggestions for Authors |
||
Comments 1: IDEA is cited as 2004, which should be noted that for the year 2040 is the IDEIA (Improvement Act)
|
||
Response 1: Thank you for pointing this out. We agree with this comment and have made the correction accordingly.
|
||
Comments 2: Oftentimes the word “this” is used, where it would be clearer for the reader to have the concept directly referred to instead of the generic word of “this.” Check throughout to help increase the clarity.
Response 2: We agree with this feedback. Please see the revised manuscript with improved clarity throughout.
|
||
Comments 3: Generally, “further” is used to connect two ideas mid-sentence, whereas “Furthermore” is used to start a sentence. Fix throughout. Response 3: We agree with this feedback. Please see the revised manuscript with these corrections.
|
||
Comments 4: Page 3 line 114, why are the children’s needs not met? Additional detail would be helpful to add here.
|
||
Response 3: Thank you for this catch. We have addressed this concern to the best of our ability on lines 131 and 132.
|
||
Comments 5: Relationship is a term usually used to define a connection between two individuals, whereas the word relation is a term used to define the connection between two variables. There are a few instances where the word relation may be a better fit over that of relationship. |
||
Response 5: The authors agree and the change has been made in the manuscript accordingly |
Comments 6: What does “advance equity” mean page 5 line 235, I think a specifier here of what this specifically means will help the reader contextualize what this looks like and shift away from an abstract goal. |
Response 6: The authors agree and have made the changes found on lines 262 and 263.
|
Comments 7: Figure 1 is very helpful and well done. There are some small errors to be corrected to help with professionalism. - Make the approach for capitalization after the bullet lists consistent - Self-awareness, self-care generally have a dash in between the words
|
Response 7: Thank you for this catch. The visual has been revised to address these errors.
|
Comments 8: Indenting spacing is off in Conclusion and Call to Action. |
Response 8: Thank you for this catch. This formatting issue has been addressed.
|
Comments 9: In the Recommendations section, for practice recommendations are there any models you would suggest for collecting data on the social-emotional areas (such as specific measures or interview protocols)? |
Response 7: Given the variation of models and instruments related to SED/SEL, the authors felt adding this to the recommendations section without adding considerable context might confuse the reader and divert from the purpose of the recommendations. |
Comments 9: The Research recommendation for “IECMHC within early intervention programs.” is a little unclear, reading it I wonder are you suggesting program evaluations of effective IECMHC supports? Add a little more detail here to clarify what you mean . |
Response 9: Thank you for this feedback. We have added language to improve the clarity of the recommendation. |
Comments 10: Part of the template is still present in “author contributions” |
Response 10: Thank you for this catch. We have removed that portion |
Comments 11: Remove the “ in the Funding Statement |
Response 11: Thank you for this catch. We have removed that portion |
Comments 12: Sometimes social-emotional has a dash, sometimes not. Please evaluate for consistency throughout. |
Response 12: Thank you for this catch. We have made the edits to maintain consistency throughout. |
Comments 13: The abbreviations may be better as a table higher up in the article, or as a supplemental material. |
Response 13: The authors agree with this recommendation and have embedded a table on line 70. |